# The C-Terminal Domain of HIV-1 Integrase: A Swiss Army Knife for the Virus?

**DOI:** 10.3390/v14071397

**Published:** 2022-06-27

**Authors:** Cecilia Rocchi, Patrice Gouet, Vincent Parissi, Francesca Fiorini

**Affiliations:** 1Molecular Microbiology and Structural Biochemistry (MMSB), CNRS, University of Lyon 1, UMR 5086, 69367 Lyon, France; cecilia.rocchi@ibcp.fr (C.R.); patrice.gouet@ibcp.fr (P.G.); 2Viral DNA Integration and Chromatin Dynamics Network (DyNAVir), 33076 Bordeaux, France; vincent.parissi@u-bordeaux.fr; 3Fundamental Microbiology and Pathogenicity (MFP), CNRS, University of Bordeaux, UMR5234, 33405 Bordeaux, France

**Keywords:** integrase, HIV-1, CTD structure, CTD-DNA interaction, integrase molecular partners

## Abstract

Retroviral integrase is a multimeric enzyme that catalyzes the integration of reverse-transcribed viral DNA into the cellular genome. Beyond integration, the Human immunodeficiency virus type 1 (HIV-1) integrase is also involved in many other steps of the viral life cycle, such as reverse transcription, nuclear import, virion morphogenesis and proviral transcription. All these additional functions seem to depend on the action of the integrase C-terminal domain (CTD) that works as a molecular hub, interacting with many different viral and cellular partners. In this review, we discuss structural issues concerning the CTD, with particular attention paid to its interaction with nucleic acids. We also provide a detailed map of post-translational modifications and interaction with molecular partners.

## 1. Introduction

Human immunodeficiency virus (HIV-1) continues to be a major public health threat worldwide. It is estimated that between 38 and 45 million people currently live with HIV-1, and no effective cure is available (World Health Organization data). Moreover, a highly virulent variant of HIV-1 (“VB” variant for “virulent subtype B”) has recently been characterized [1].

HIV-1 belongs to the lentivirus genus of the Retroviridae family that requires the integration of a DNA copy of its RNA genome (gRNA) into the genome of the infected cell. To do this, retroviruses carry the integrase (IN), the enzyme responsible for this insertion, within the virions. IN is a multimeric enzyme that associates with viral DNA (vDNA) in a nucleoprotein complex termed intasome, which targets cellular DNA (tDNA). IN catalyses two consecutive reactions: first, the 3′ processing, where the two nucleotides at the 3′ ends of the viral DNA are excised to leave an invariant 3′-CA dinucleotide protruding end, and second, the DNA strand-transfer reaction, where the activated 3′ ends attack the chromosomal DNA and joint it covalently (reviewed in [2]). The process is assumed to be completed by the host cell’s DNA repair enzymes, whose nature remains mostly unknown (Figure 1).

Beyond integration, HIV-1 IN possesses accessory functions that were recently highlighted [3,4,5,6,7,8]. These integration-independent functions are supported by evidence of several amino acid substitutions in IN—referred to as class-II mutations—affecting virion assembly, morphogenesis and reverse transcription in target cells, and in some instances without impacting IN catalytic function (reviewed in [7,9,10,11]). Many of these functions of IN seem to be associated with its C-terminal domain (CTD), which works as the “Swiss army knife” of HIV-1 integrase. We will focus our review on these IN CTD-associated functions, which are often essential for viral replication.

## 2. Integrase within the Intasome: Conserved Core Organization

More than 40 years ago, IN was enzymatically characterized [13], as reviewed in [14], and numerous biochemical and structural studies followed to clarify a variety of mechanistic aspects [2,15]. IN is 288 amino acids (aas) long in HIV-1 and consists of three structured domains: the N-terminal domain (NTD, residues 1–47 aa), the catalytic core domain (CCD, residues 59–202) and the CTD (residues 220–270) [16]. The NTD is involved in enzyme multimerization and shows a His2Cys2 motif coordinating a Zn2+ ion [17,18,19]. The CCD harbours the essential catalytic triad Asp, Asp, Glu (D,D,E) that coordinates two Mg2+ cofactors and folds similarly to nucleotidyl-transferases and nucleases [20]. The CTD is also involved in DNA interaction, multimerization and several other functions (see below). The three domains are interspaced by two linkers (residues 48–58 and 203–219, respectively), and the CTD is followed by a flexible tail of 18 residues (residues 270–288) (Figure 2A). Interestingly, the IN from the highly virulent HIV-1 VB variant presents five nonconservative mutations distributed throughout the protein (S23G, G58E, T123A, A195P and S282G) [1].

Interdomain flexibility has hindered the structural study of the full-length IN for years, and structures of individual domains from different IN retroviral species were first determined during the 1990s [2]. The structures of NTD-CCD and CCD-CTD, two-domain fragments, were determined in turn, including those of HIV-1 IN [21,22], until structural biologists took advantage of the stabilization offered to IN within the intasome to solve its full-length structure [23,24]. The first structure of a complete retroviral intasome to be solved was that of prototype foamy virus (PFV) with and without tDNA (Figure 2B) [25,26,27]. PFV intasome reveals a tetrameric elongated assembly for IN, with a 2-fold symmetry in which the inner monomers are organized around vDNA and bring 3′ extremities close to tDNA (Figure 2B). The inner CTD, also termed the synaptic CTD, lies between the inner NTD and the inner CCD. The two symmetry-related inner CTDs flank the vDNA and have no contact with each other, but participate in the distortion of the tDNA in association with the catalytic inner CCD to make it cleavable ([27] Figure 2B). The outer CCD has a dimeric interface with the inner CCD and is noncatalytic (Figure 2B).

Lentiviral intasomes are highly heterogeneous in size and can vary between tetramers, dodecamers, hexadecamers and proto-intasome stacks [23,24,28,29]. Although the functional impact of intasome multimerization is intensely debated, the presence of a PFV-like conserved intasome core (CIC) with a synaptic CTD is the consensus.

## 3. The Structure of HIV-1 IN CTD, Interdomains and DNA Interactions

In this review, we aim to recapitulate all the key issues around the roles of HIV-1 IN CTD, which have been hotly debated since its first characterization [31,32].

The first structure of HIV-1 IN CTD, spanning from residues 220 to 270, was determined by nuclear magnetic resonance spectroscopy (NMR) and is dimeric [33]. The 18aa-long flexible C-terminal tail was removed to increase solubility. Each monomer displays a SH3-like β-barrel fold, with five beta strands assembled in two tightly packed antiparallel β-sheets (Figure 3A). SH3-like domains are protein–protein interaction domains that can perform a great variety of functions by modulating their loops and electrostatic motifs. For example, classical SH3 domains are able to bind proline-rich motifs to their target partners through a hydrophobic pocket [34]. On the other hand, Tudor domains, which also belong to the SH3-like superfamily, recognize mono- or poly-methylated arginines or lysines of their interactors to mediate their functions in RNA metabolism, recruit effectors during DNA damage response or promote chromatin activation or silencing [35]. Chromodomains are also characterized by a SH3-like barrel and are known to be associated with chromatin remodelling via their specific binding to modified histone tails [36].

HIV-1 IN CTD220-270 shows nonspecific DNA-binding properties and contains many basic residues [31,32,39,40,41]. The dimer exhibits a saddle-shaped groove delineated by long loop connecting strands β1 and β2 (Figure 3A). Earlier mutational studies suggested that Lys 264 (K264), located on the edge of the groove, was involved in DNA binding [31], and the entire groove itself with its concave form and protruding residues R231, K258 and K264 was supposed at the time to accommodate double-stranded DNA [33]. Yet, recent intasome structures have discarded this hypothesis, as described hereafter, and a molecular electrostatic surface reveals the presence of a basic patch that is adjacent to the saddle-shaped groove. This patch comprises residues from the end of β1, the end of β2, the loop β4-β5 and strand β5 (R228, K244, R262, R263, K264, K266, R269; Figure 3A).

The dimeric interface is perpendicular to the bottom of the groove and is mainly hydrophobic. There, the antiparallel sheet formed by β2, β3 and β4 stacks with its symmetric and buries the Trp 243 (W243) residue which faces Leu 242 (L242) and Ile 257 (I257) of the second subunit (Figure 3B). This interface is a structural hallmark of the HIV-1 IN CTD fragment in solution [33,42,43]. One exception is the crystal structure of a trimeric CTD harbouring an N-terminal polyhistidine-tag that coordinated Ni2+ ions along a 3-fold molecular axis [37] (Figure 3C). The W243 is not accessible within this trimer, impeding the formation of any canonical dimeric interface in the crystal.

The two CTDs do not interact with each other in the dimeric structure of the CCD-CTD two-domain fragment. They are at the extremity of a Y-shape dimer formed by the CCD-CCD core and two helical linkers (PDB: 1EX4, [38]) (Figure 3D). Notably, this rigid Y-shape dimer conformation is also observed in the cryo-EM structure of the dodecameric cleaved synaptic complex (CSC, i.e., IN with cleaved vDNA; Figure 1 and Figure 4A) intasome of HIV-1 [23]. At atomic resolution, this structure with 2-fold symmetry shows two pairs of synaptic CTDs organized around the vDNA (Figure 4A; synaptic and synaptic–distal CTDs), while the other two pairs are away from the active site (Figure 4A; distal CTDs). The functional plasticity of the CTD can be appreciated in this oligomeric form with various protein–protein and protein–DNA interfaces (Figure 4B(a, b, and c); [23]). For example, the synaptic–distal CTD interacts on one side with the synaptic CTD (Figure 4B(a)) and forms a canonical CTD:CTD dimeric interface with a distal CTD on another side (Figure 4B(b)), reconciling the structural information obtained so far [23,33].

Focusing on interactions with vDNA, an interesting role is played by the basic patch of the synaptic CTD, which lies along the vDNA phosphate backbone, with the R263 residue making the closest contacts (Figure 3A and Figure 5A). The flexible loop β1-β2 is also an important nucleic-acid-interacting site with its highly accessible R231 residue pointing toward the catalytically competent 3′-cytosine in the synaptic DNA (Figure 5A). Consistently, in the STC structure, this arginine directly interacts with a base of tDNA, contributing to its distortion [24]. Previous mutational studies have highlighted the importance of this residue for strand transfer [44].

Of interest, the 271–276 amino-terminal extremity of the flexible tail 271–288 can be observed for the first time in the CSC structure (Figure 5B). In the synaptic–distal CTD, this segment leans on an adjacent helical CCD-CTD linker and similarly folds into a helical structure (Figure 5B). This flexible CTD tail ensures several functions essential for infectivity, and pleiotropic effects are caused by some mutations, which are classified as class-II [7,10,11,45,46].

## 4. Post-Translational Modifications of HIV-1 IN-CTD

During infection, HIV-1 IN undergoes various post-translational modifications (PTMs), which tightly regulate its enzymatic function and its interaction with numerous cellular cofactors [47,48,49,50,51,52]. These modifications are mainly acetylation, ubiquitination, SUMOylation and phosphorylation and mostly occur in the C-terminal region of the protein (reviewed in [52]; Figure 6).

Several histone acetyl transferases (HATs) have been described to interact with, and modify, HIV-1 IN-CTD [47,49]. This family of enzymes is known to acetylate specific lysine residues within the histone N-terminal tail region [53], facilitating the access of the transcription machinery to DNA, as well as providing a platform for the recruitment of regulatory factors [54]. The p300 acetyl-transferase, one of the major representatives of the HAT family, has been shown to interact with IN-CTD and acetylates two lysine residues of the basic patch (K264 and K266) and a lysine of the flexible tail (K273; [47,55] (Figure 6). Another cellular HAT, GCN5, has been shown to acetylate IN-CTD on one additional lysine residue at position 258 compared to p300-acetylated residues [47,49]. Acetylation increased the IN affinity for DNA and enhanced the strand transfer activity in vitro [47]. Mutations of K264, K266 and K273 abolished HIV-1 replication, while K258 mutation did not play a significant role during infection [49].

More than a dozen cellular interactors have been identified for acetylated IN [56]. Among them, several factors involved in transcription regulation, chromatin remodelling, translation and nuclear import–export were found [56]. One important identified hit among these partners is LEDGF/p75, which is the IN interactor that has been the most studied. LEDGF/p75 is required for efficient strand transfer activity and is involved in tethering the enzyme to transcriptionally active regions of host chromosomes [56,57,58,59]. Although the binding of LEDGF/p75 primarily involves the IN CCD region, one study has described that acetylation of the IN CTD increases the interaction of the full-length IN protein with LEDGF/p75, through a still undefined mechanism [56,58,60].

SUMOylation (covalent addition of small ubiquitin-related protein) of IN occurs on three different sites, and one of them, K244, belongs to the CTD (Figure 6). These SUMOylations occur both in vitro and ex vivo, and mutations of any of the SUMOylation sites reduce HIV-1 infectivity and affect its replication at early stages before integration occurs [50]. SUMO1 and SUMO2 proteins also interact with IN, but only a portion of them bind IN covalently in the cell, likely by the action of the E2-conjugating enzyme Ubc9 [61]. Because SUMOylation-impaired IN retained wild-type catalytic activity, Zamborlini and coworkers argued that SUMOylation might participate in the modulation of the HIV-1 IN interaction network, likely regulating its affinity for cofactors [50]. Recently, three putative SUMO-interacting motifs (SIMs) have been identified along the primary sequence of HIV-1 IN [51]. SIMs allow the noncovalent interaction of human SUMO or SUMO-modified proteins, inducing different functional consequences, such as protein modification, protein localization or interaction with cofactors (reviewed in [62]). Globally, mutations of the three SIMs have several consequences, reducing IN SUMOylation and LEDGF/p75 binding [51]. IN-CTD harbours one of these SIMs (SIM3), spanning from amino acid 257 to 260 (IKVV), which directly binds SUMO3 protein (Figure 6). SIM3 is involved in viral reverse transcription, nuclear import and integration reaction [63].

The stability of HIV-1 IN is controlled by the ubiquitin–proteasome pathway. The von Hippel–Lindau binding protein 1(VBP1) directly interacts with the IN catalytic domain, inducing its ubiquitinylation by the action of the cullin2 (Cul2)-based von Hippel–Lindau (VHL) ubiquitin ligase complex [64]. This PTM occurs on the helical linker region between CCD and CTD (K211, K215, K219), and on the C-terminal tail (K273; (Figure 6) and affects IN stability at the post-integration level [64].

Lastly, the only phosphorylation documented for IN-CTD occurs at Serine 255 (S255) (Figure 6) and is operated by the serine–threonine kinase GCN2 (general control nonderepressible 2) [65]. This kinase is activated upon HIV-1 infection and restricts viral integration and replication by directly modifying IN [65,66,67]. GCN2 has been proposed to represent a defence mechanism of the cell instead of an integrase partner.

## 5. Interaction of IN-CTD with Reverse Transcriptase

The reverse transcription of gRNA in vDNA is an essential step of the retroviral life cycle and is mediated by the reverse transcriptase (RT), a viral enzyme, which, together with IN, is incorporated into virions. The RT reaction converts single-stranded RNA into double-stranded DNA that is subsequently integrated by IN into the host genome within the nucleus. While the spatiotemporality of the reverse transcription is still under debate, recent data suggest that this reaction may occur during the translocation of the cone-shaped viral core, which contains the RT- ribonucleoprotein complex (RT-RNP), from the cytoplasm to the nucleus of the newly infected cell [68]. The viral DNA produced by this reaction is then associated with pre-integration complexes (PICs), also containing IN and several other viral proteins. Both individual subunits of RT, p66 and p55, physically interact with IN-CTD independently through the presence of nucleic acid [69]. The residues involved in RT:IN interaction have been precisely mapped in the presence and absence of DNA (Figure 6) [70,71]. Interestingly, the presence of DNA bound to RT expands the RT-binding surface on IN [70,71]. The RT:IN complex formation is functionally important because the presence of IN stimulates both the initiation and the elongation of reverse transcription in vitro [3]. Mutants in RT-binding IN residues significantly impair viral replication at the reverse transcription level ex vivo [10,70,71]. Importantly, the presence of IN within the RT-RNP is necessary for efficient initiation of viral DNA synthesis [3,10,69,70,71,72,73].

The packaging of tRNALys3 from the host cytoplasm into the new viral particles is one of the most essential steps to initiating reverse transcription in the virions. For this reaction to occur, the mitochondrial lysyl-tRNA synthetase (mLysRS), an enzyme that catalyzes aminoacylation of tRNALys3, has to be incorporated into the virions [74]. It has been shown that HIV-1 IN, within the Gag or GagPol polyprotein, is responsible for the encapsidation of mLysRS and tRNALys3 [4,5]. In particular, the IN CTD is responsible for the interaction with a dimer of mLysRS [75].

## 6. Role of the IN-CTD during Nuclear Import

Recent high resolution tomography imaging of HIV-1-infected cells has revealed that an intact viral capsid can penetrate the nuclear pore complex (NPC), disassemble in the nucleoplasm and release the retrotranscribed vDNA within the PIC near the host genome [68]. The morphology of the capsid undergoes dynamic remodelling during the travel from the cytoplasm to the nucleoplasm, influencing the integration sites [68,83]. Past evidence indicated that IN is a major viral determinant in HIV-1 nuclear import [6], especially its critical interaction with the cellular transportin-SR2 (TRN-SR2) (reviewed in [84]). TRN-SR2 strongly interacts with several charged residues of IN CTD (R262, R263 and K264; Figure 6, [80,81]). The arginine 263 and lysine 264 of IN-CTD also interact with Importin α3, another cellular nuclear import factor [77,85]. Two other Importines, α1 and 7, interact with IN but also with other residues, although the interacting site of importin 7 partially overlaps with that of Importine α3 and TRN-SR2 (Figure 6) [6,77,78,79].

## 7. Integrase and Chromatin Remodelling System

The interaction of HIV-1 IN with specific host factors is responsible for the integration site selection within the cellular genome [86,87]. Interestingly, those integration sites positively correlate with sites showing histone modifications characteristic of actively transcribed chromatin [88]. As mentioned before, a crucial role is played by the LEDGF/p75 cofactor that tightly binds chromatin, engaging the histone H3 tail containing trimethylated Lys36 (H3K36me3) and the DNA through its N-terminal domain, and binds IN through its C-terminal IBD (integrase-binding domain), [89,90,91]. Moreover, LEDGF/p75 significantly stimulated the strand transfer activity of HIV-1 IN, playing a direct role in HIV-1 integration [92]. Interestingly, IN must also interact with the nucleosome for efficient integration [76]. The interaction of HIV-1 IN with the N-terminal tail region of H4 histone has been shown in vitro, and the increasing efficiency of DNA integration within mononucleosomes has also been observed [76]. IN-CTD has been suggested to be responsible for the interaction with the tail of H4. In particular, an in silico blind docking simulation and NMR approaches identified that the saddle-shaped groove surface comprising residues R231 and K258 could accommodate the H4 histone tail [76,93]. Analysis of IN-CTD mutants within this groove revealed that, among all the residues, R231 seems to be the most specific for H4 tail binding (Figure 6) and mediates the interaction of IN with mononucleosomes [76]. Very recently, LEDGF/p75 has been shown to modulate this IN-intrinsic chromatin-binding property in a chromosome spreads model [94], suggesting a close interplay between the IN CTD function and the cellular factor.

HIV-1 IN also interacts with INI1, a subunit of the SWI/SNF chromatin remodelling complex [95]. INI1 is selectively incorporated into HIV-1 particles with IN and seems to act at early and late stages of the HIV-1 cycle by affecting virion assembly, reverse transcription and integration [96,97,98,99]. In particular, integration activity is directly influenced by the presence of INI1, and the addition of the SWI/SNF complex enhances integration into nucleosomal targets in vitro [95,100,101]. This integration into chromatinized substrates does not depend on the mere presence of INI1, but is strictly dependent on the chromatin remodelling activity of SWI/SNF [101]. Moreover, it has been shown that INI1 inhibits a 3′ processing reaction of vDNA by the IN:LEDGF complex [102]. A docking study supported by biochemical analysis suggested that the conserved Rpt1 domain of INI1 interacts with IN-CTD by hydrophobic and specific ionic interactions [82]. In particular, the tryptophane 235 of IN CTD (W235E), especially its aromatic nature, is important for the binding to INI1 [82] (Figure 6).

## 8. RNA-Binding Properties of IN and Impact on Virion Morphogenesis

During HIV-1 virion morphogenesis, the conical capsid assembled around the viral ribonucleoprotein complex (vRNP) mainly composed of two copies of genomic viral RNA coated by nucleocapsid protein (NC) and replication enzymes such as IN and RT. Some class-II IN mutants and IN inhibitor-treated viruses showed a particular phenotype named “eccentric condensate”, in which vRNP nucleates outside the capsid lattice [7,103,104,105,106,107,108]. The same phenotype was also observed in IN-deleted virions [109]. In 2016, Kvaratskhelia and coworkers showed for the first time the interaction of HIV-1 IN with viral genomic RNA (gRNA) ([7], reviewed in [110]). IN prefers to bind the discrete structured portion of gRNA, and deletions of these structural elements reduced the IN binding affinity [7]. A specific disruption of this interaction produced eccentric condensate virions, revealing a link between the IN:gRNA binding and HIV-1 morphogenesis [7,107]. In this pivotal study, only three lysine residues, K264, K266 and K273, belonging to the CTD and its tail, have been identified to interact with gRNA [7]. These residues were previously mentioned in Section 4 and can be acetylated. Analysis of K264A/K266A and R269A/K273A double mutants revealed that both proteins efficiently bind LEDGF/p75, while K264A/K266A substitutions exclusively affect integration activity and R269A/K273A prevent RNA binding without significantly affecting other known functions of the enzyme in vitro [7]. A critical role in RNA binding seems to be played by the tetramerization ability of HIV-1 IN [7,46]. The majority of the class-II IN mutants form dimers and monomers almost exclusively, dramatically decreasing the RNA affinity ex vivo and in vitro [7,46]. In contrast, the double mutations K264A/K266A and R269A/K273A directly affect the RNA interaction without perturbing the multimerization ability of IN [7,46].

The consequence of eccentric condensate virions is the inability to complete reverse transcription in infected cells due to a premature degradation of gRNA [105,107]. Most likely, the mislocalization outside the HIV-1 core causes the gRNA to lose the physical protection exerted by the capsid, inducing the rapid degradation of RNA and blocking the reverse transcription [105,107]. Notably, the same eccentric phenotype associated with an early reverse transcription blockage has been observed in mutants of the capsid protein that destabilized the capsid core [111,112,113,114].

## 9. HIV-1 IN in Proviral Transcription

Crosslinking-immuno-precipitation sequencing (CLIP-seq) experiments have identified the trans-activation response element RNA (TAR) located in the HIV-1 5′-untranslated region (UTR) as the main binding site for tetrameric IN [7,46]. TAR is one of the most conserved RNA sequences in HIV-1. It is a noncoding RNA which plays a critical role during proviral transcription (reviewed in [115]). During this process, the cellular RNA Polymerase II (Pol II) stalled shortly after initiation of transcription due to the presence of negative elongation factors as well as nucleosomes downstream the transcription start site [116,117]. The nascent TAR RNA boosts the transcription elongation by binding viral transcriptional trans-activator Tat protein that hijacks the cellular super elongation complex to the advantage of provirus [118,119,120,121]. This complex triggers a cascade of phosphorylation which activates Pol II elongation activity and recruits positive chromatin remodellers [63]; reviewed in [122].

TAR RNA is 59 nucleotides long and is present in all classes of viral transcripts. It features a long stem terminating in an apical structure composed of a three-nucleotide bulge followed by four base-pairs stem and a terminal hexaloop (Figure 7A). The structure of TAR is highly dynamic and adopts a specific conformation once bound to a ligand (reviewed in [115], Figure 7B). The presence of the bulge and the loop in TAR RNA, rather than its sequence, is critical for IN binding, as the deletion of one or both markedly decreases the binding affinity of full-length protein [7]. A crosslinking-coupled SHAPE assay using the IN:LEDGF/p75 IBD complex and the whole HIV-1 5′UTR revealed a crosslinking site located on C39, near the bulge region of TAR [123] (Figure 7A).

The direct interaction of HIV-1 IN CTD with TAR has also been assessed (https://doi.org/10.1101/2021.10.21.465253, accessed on 31 May 2022). Interestingly, the tail region encompassing the last 18 residues interacts with the guanine G34 of the apical hexaloop (https://doi.org/10.1101/2021.10.21.465253, accessed on 31 May 2022; Figure 7A,B). This interaction allows the tail region to act as a sensor of the proper TAR structure (https://doi.org/10.1101/2021.10.21.465253, accessed on 31 May 2022). Intriguingly, the binding of IN induces conformational changes of a TAR apical hexaloop overexposing a critical nucleotide (https://doi.org/10.1101/2021.10.21.465253, accessed on 31 May 2022; [123]). A similar conformation of TAR has been observed in the structure of the TAR:Tat:SEC –core complex, allowing the binding of Cyclin T1 to TAR for a proviral transcription boost [120].

The link between IN-CTD and proviral transcription has been recently highlighted for HIV-1 [8]. In this work, the authors mutated basic residues of the saddle-shaped groove (K258), the basic patch (K264, K266) and the tail (K273) undergoing PTM, with the aim of affecting only IN acetylation without changing the charge and size of the residue [8]. Whether this acetylation is important for HIV-1 integration ex vivo is still a subject of debate [47,55]. The simultaneous conservative mutation of all acetylated residues affects the transcription of proviral genes shortly after infection, while it has only modest effects on early steps of HIV-1 replication, such as reverse transcription and integration [8]. Interestingly, some of these residues mediate the interaction of IN with TAR RNA [7,46]. The effect on RNA binding of conservative mutations of IN CTD as well as the effect of nonconservative mutations on proviral transcription are unknown so far.

The ex vivo data on the intervention of IN in proviral transcription are supported by in vitro analysis of the interplay between IN and Tat for TAR RNA binding (https://doi.org/10.1101/2021.10.21.465253, accessed on 31 May 2022). The modification of TAR structure by IN-CTD binding primes the subsequent binding of Tat, which finally displaces IN from TAR (https://doi.org/10.1101/2021.10.21.465253, accessed on 31 May 2022).

Interestingly, these CTD mutations also affect the half-life of IN binding to the integrated DNA, suggesting that the enzyme could play an active role during the very early post-integrative events, such as DNA repair of the insertion site [8].

## 10. Conclusions

Beyond its essential function in vDNA integration, IN plays a role in many other steps of the HIV-1 life cycle. Consistent with its availability within the intasome, the CTD of IN appears to represent a molecular hub of these accessory functions, undergoing most of the PTMs and mediating several fundamental interactions with molecular partners.

Much remains to be done to study the role of IN in post-integration events and its exact role during virion morphogenesis. Understanding whether IN is somehow involved in DNA repair of integration sites or exactly how it supports the early stage of proviral transcription remain open questions. Structural information would be particularly needed to better understand the underlying molecular mechanisms behind these new roles of IN and the involvement of IN CTD in these new functions.

## Figures and Tables

**Figure 1 viruses-14-01397-f001:**
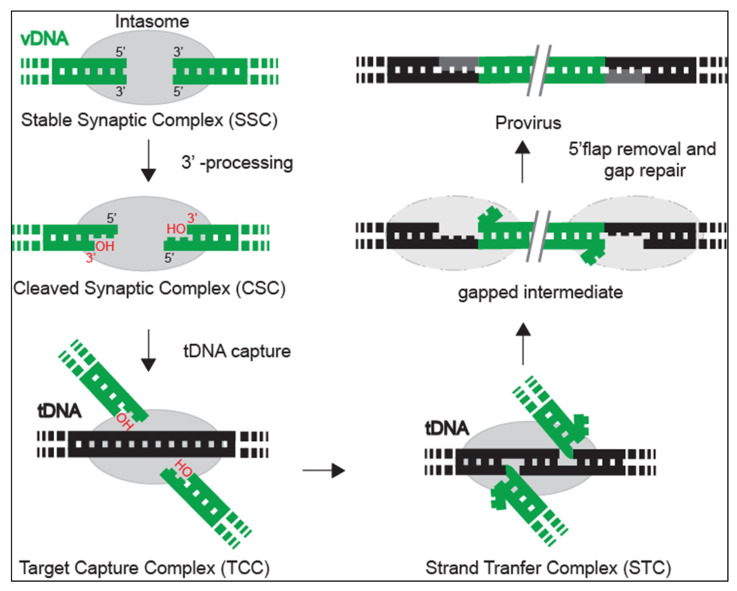
Integrase catalytic functions and intasome complexes. The integration process starts when IN (depicted by grey oval) associates with two ends of vDNA (in green), establishing a stable synaptic complex (SSC), which is the first intermediate of the integration reaction pathway. Then, IN cleaves two nucleotides at each 3′ end of vDNA, forming the cleaved synaptic complex (CSC). Then, the tDNA (in black) is captured to form the target capture complex (TCC), and the integration occurs within the strand transfer complex (STC). Once the STC disassembles, the two flapping nucleotides from 5′ are removed, and the gapped intermediate is repaired by cellular enzymes to fill the single strand connection between vDNA and tDNA of the resulting provirus [12].

**Figure 2 viruses-14-01397-f002:**
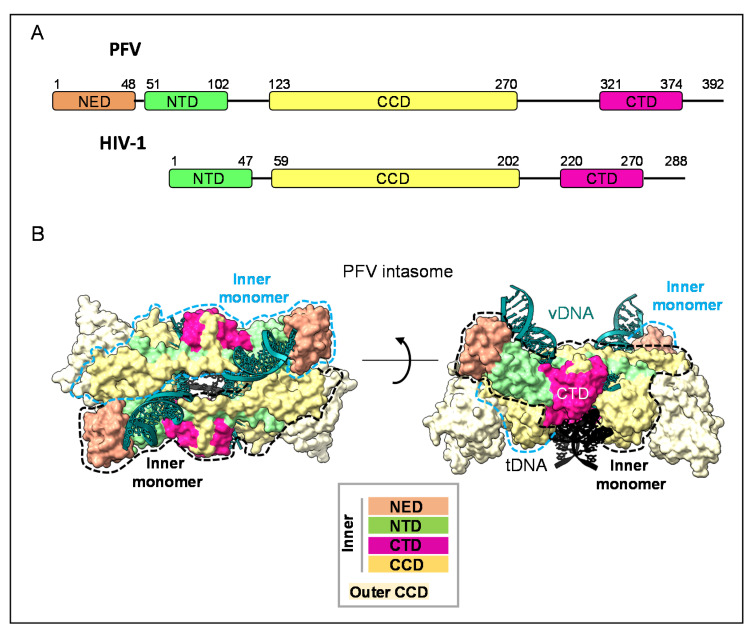
Domain organization of IN and structure of PFV intasome. (**A**). Integrase (IN) domains and interdomains are indicated by boxes and lines, respectively. Starting and ending amino acid residues are indicated above the boxes [30]. PFV integrase (upper representation) has an additional N-terminal extension domain (NED, in orange) that is not present in HIV-1 integrase (lower representation). N-terminal domain (NTD) is in green, the catalytic core domain (CCD) in yellow and C-terminal domain (CTD) in dark pink. This domain organization is conserved among retroviral integrases [2]. (**B**) Surface representation of the crystal structure (PDB: 3OS0, [27]) of PFV strand-transfer complex (STC) viewed along and perpendicular to the two-fold axis. IN is organized as an elongated tetramer along the DNA; the inner IN monomers are organized around vDNA and bring 3′ extremities close to tDNA. The two inner monomers are outlined by dashed lines (light blue and black). The vDNA, tDNA and IN domains are indicated and follow the colour code of Figure 2A. Outer monomers are represented only by outer noncatalytic CCD domains, which are coloured in pale yellow.

**Figure 3 viruses-14-01397-f003:**
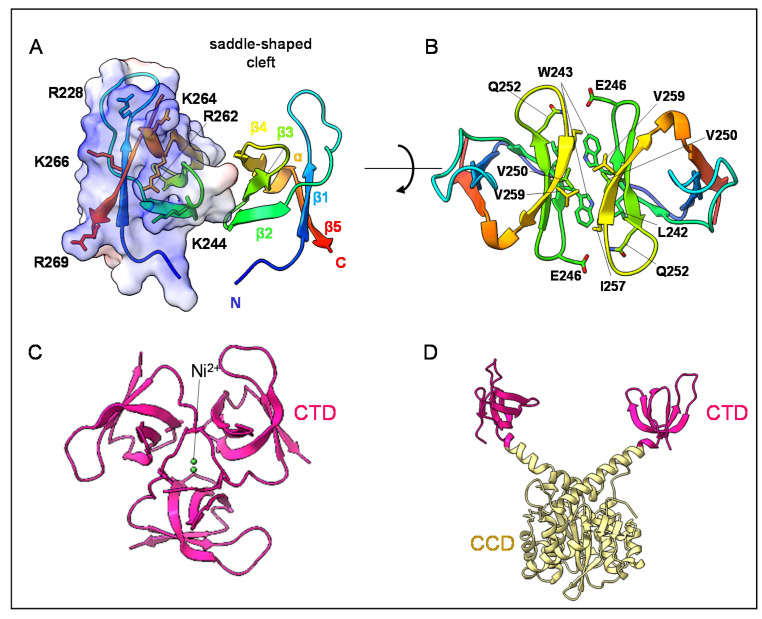
Oligomeric forms of HIV-1 CTD. (**A**). HIV-1 CTD dimer in solution (PDB: 1IHV; [33]). The surface representation of the right monomer of the CTD shows its electrostatic charge distribution. A basic patch is observed (R228, K244, R262, K264, K266, R269) near a saddle-shaped groove (R231, K258), indicated with a dashed line (grey). The left monomer is shown in a cartoon depiction and coloured in rainbow colours from the N- to C-terminus in order to emphasize the SH3-like fold. (**B**). Interacting residues at the canonical CTD-CTD dimeric interface are shown by lines and labelled (L242, W243, E246, Q252, I257). (**C**). Crystal structure of a trimeric CTD (PDB: 6T6J; [37]) harbouring an N-terminal polyhistidine-tag that coordinates Ni2+ ions (in green) along a three-fold molecular axis. (**D**). HIV-1 two-domain CCD-CTD crystal structure (PDB: 1EX4, [21,38]). The CCD-CTD protein fragment acquires a Y-shape dimeric conformation. The CCDs dimerized, and the respective CTDs (in dark pink) are at the end of the two helical CCD linkers (in pale yellow).

**Figure 4 viruses-14-01397-f004:**
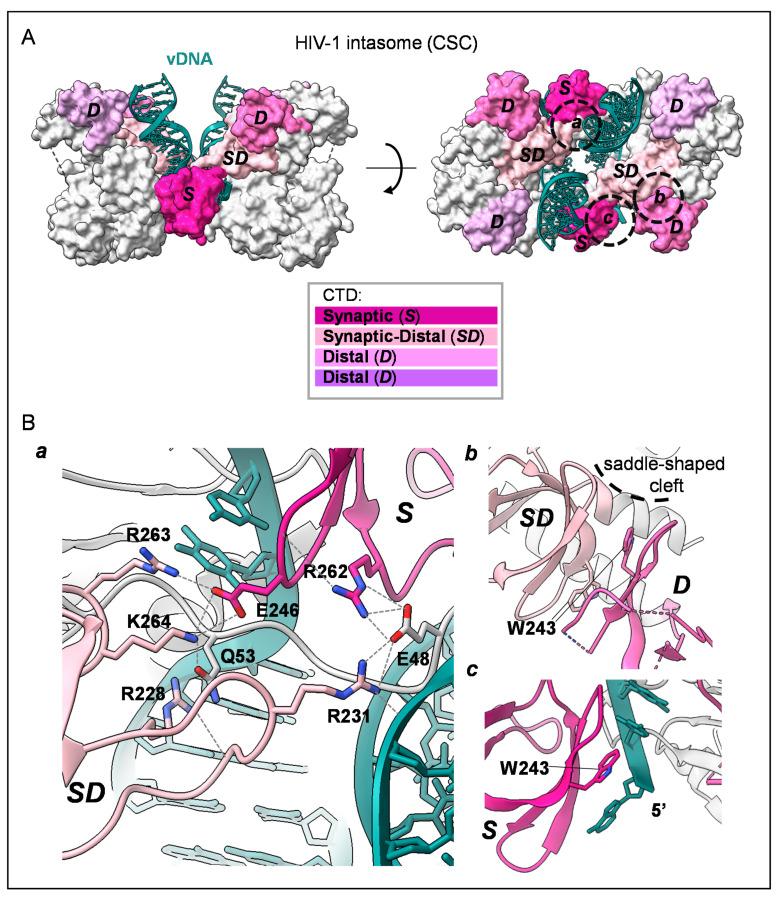
Localization of IN CTD within the HIV-1 cleaved synaptic complex (CSC). (**A**). Surface representation of HIV-1 CSC: only CTDs and vDNA are coloured with four different shades of pink (PDB: 6PUT, [23]). Synaptic CTD (S, dark pink) and synaptic–distal (SD, in light pink) are assembled around vDNA (dark cyan), while distal CTDs (in violet and lilac) are not in contact with vDNA. (**B**). Focus on a, b and c circles of (**a**) interactions between R228, R231, R263 and K264 of distal–synaptic CTD (SD, in light pink), Q53 and E48 of NTD-CCD linker (in grey) and E246 and R262 of synaptic CTD (S, in dark pink); (**b**) dimeric interface between distal–synaptic and distal CTDs: the two W243s comprise the canonical CTD:CTD dimeric interface and are indicated by lines while the saddle-shaped cleft is highlighted with the curved dashed line; (**c**) illustration of the plasticity of the CTD: in the synaptic CTD, W243 interacts with the 5′ end of vDNA.

**Figure 5 viruses-14-01397-f005:**
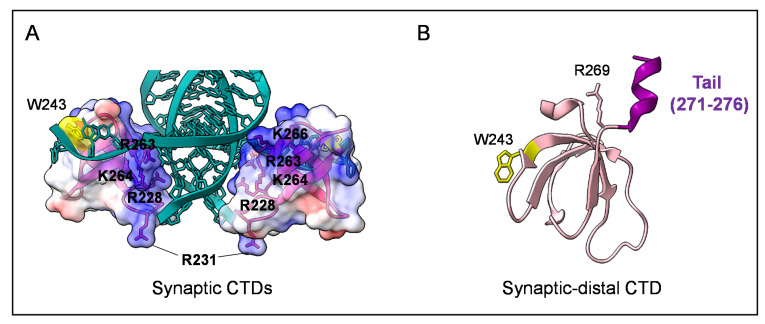
Interaction of synaptic CTDs with DNA and structure of synaptic––distal CTD. (**A**). Synaptic CTDs within CSC structure (PDB: 6PUT, [23]). Electrostatic surfaces show the position of the basic patch oriented towards the vDNA (residues are labelled and indicated with lines, W243 is highlighted in yellow). Of particular interest, R231 is oriented on the opposite side of the vDNA and towards the cleft, where in the next step of integration pathway, the cleaved tDNA will establish the TCC and STC conformations. This orientation is consistent with the 5U1C structure of the HIV–-1 STC, where R231 is a specific contact with a tDNA base [23,24]. (**B**). Cartoon representation of synaptic–distal CTD (SD) within CSC structure (other IN domains and vDNA are not shown); it importantly shows the most complete version of the C–-terminal tail (until A276) folding into a helical structure (in purple) (PDB: 6PUT, [23]).

**Figure 6 viruses-14-01397-f006:**
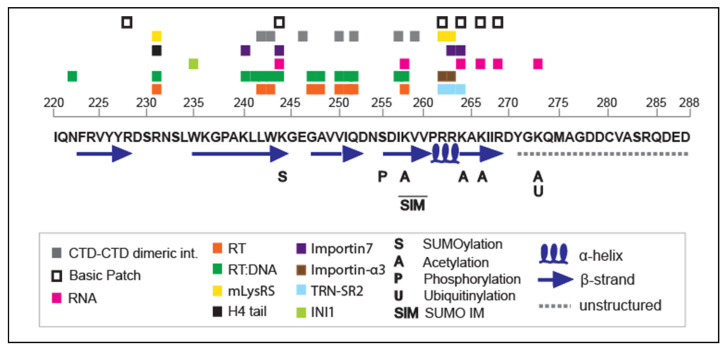
Scheme indicating the IN CTD amino acid residues involved in various interactions and post-translational modifications. The amino numbering is indicated above the IN CTD sequence, between aa 220 and 288. The coloured squares indicate the residues involved in the interaction of IN CTD with: RNA in dark pink [7,46]; RT and RT:DNA complex in orange and dark green [70,71]; Human Mitochondrial Lysyl-tRNA Synthetase in yellow [75]; histone H4 tail in black [76]; Importin 7 and α3 in violet and brown, respectively [6,77,78,79]; TRN-SR2 in light blue [80,81]; and INI1 in light green [82]. The residues involved in CTD-CTD dimerization are labelled with grey squares, and the residues belonging to the basic patch with empty squares. Several post-translational modifications are labelled: S is for SUMOylation [50]; A for acetylation [47,49,55]; P for phosphorylation [65]; U for ubiquitination [64] and SIM for SUMO-interacting motif [63]. Elements of secondary structure are also represented at the bottom of the sequence.

**Figure 7 viruses-14-01397-f007:**
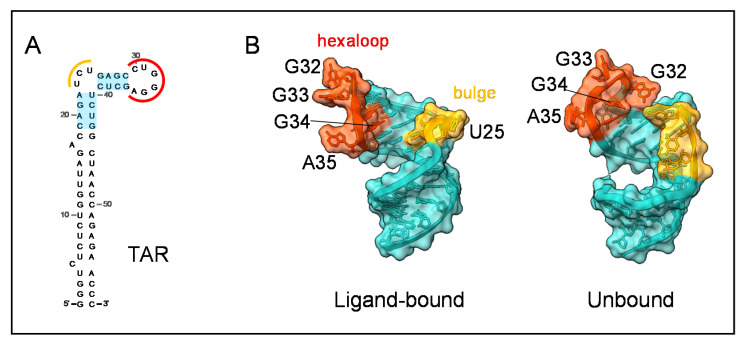
Nucleotide sequence and secondary structure of HIV-1 TAR RNA. (**A**). Highlighted in light blue are the stems present in the solved structures, in yellow the UCU bulge and in orange the G-rich hexaloop. (**B**). Surface representation of ligand-bound state (PDB: 6MCE, TAR bound to the arginine-rich RNA-binding region of Tat) and unbound state of TAR RNA (PDB: 1ANR, model 1). Indicated bases of the G residues of the hexaloop and the flanking UCU bulge are oriented inward, contributing to the bend of the helical axis and the tightening of TAR major groove (adapted from [115]).

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
