# Peer review of "The C-Terminal Domain of HIV-1 Integrase: A Swiss Army Knife for the Virus?"

_viruses, 2022, doi:10.3390/v14071397_

Round 1

Reviewer 1 Report

1)     lines 23-28: In the first paragraph of the “Introduction “, authors described about a highly virulent variant of HIV-1. Most of readers may be interested in whether IN mutations are associated with the emergence of virulent variants? If so, please add description for possible contribution of IN mutations in the text.

2)     lines 215-218: Is interaction with LEDGF/p75 CTD-mediated?

Fig. 7 : Please clarify the ligand for TAR in Fig. 7B, Tat or IN-CTD?

Author Response

We thank this reviewer for his/her work and comments.

1) lines 23-28: In the first paragraph of the “Introduction “, authors described about a highly virulent variant of HIV-1. Most of readers may be interested in whether IN mutations are associated with the emergence of virulent variants? If so, please add description for possible contribution of IN mutations in the text.

In the paragraph n 2, line 68, we have added the sentence “Interestingly, the IN from the highly virulent HIV-1 VB variant presents five non-conservative mutations distributed throughout the protein (S23G, G58E, T123A, A195P and S282G).”

We have also presented the new HIV-1 variant as “VB” variant for “virulent subtype B” in line 27.

2) lines 215-218: Is interaction with LEDGF/p75 CTD-mediated? 

This is a good point. The direct interaction between LDGF/p75 and IN CTD has not been observed. LEDGF/p75 binds two regions of IN: the first is between residues W131 and W132 and the second from I161 to E170 within the CCD (Busschots et al 2007). However, Allouch and Ceresto in 2011 observed that fully acetylated IN precipitated more LEDGF/p75 in a special two-hybrid experiment, suggesting that the IN CTD may also be, directly or indirectly, involved.

We have added the sentence “Although the binding of LEDGF/p75 primarily involves the IN CCD region, one study has described that acetylation of the IN CTD increases the interaction of the full-length IN protein with LEDGF/p75, through a yet undefined mechanism”

Fig. 7 : Please clarify the ligand for TAR in Fig. 7B, Tat or IN-CTD?

We specified in the legend “TAR bound to the Arginine-rich RNA-binding region of Tat”

This conformation of TAR has also been observed with other ligands such as a lab-evolved cyclic peptide mimicking Tat ARM: for this reason, we prefer to keep the generic term “ligand-bound” in the figure.  However, if Referee 1 deems it more appropriate to use another term, we can change it.

Reviewer 2 Report

Excellent and well written review that goes into great detail explaining the multi-functional role of CTD of IN. This should be a lead review for this region of IN and help the field explore questions that remained to be answered. Figures are excellent and very complementary to what is described. A small concluding paragraph that summarizes findings and important questions to be answered should be included. Build off the last paragraph or use that as a start. Minor edits are as followed:

Line 24: should be threats

Lines 26-28: not sure if this necessary

Line 36: should be ends

Line 37: should be joins

Line 40: highlighted needs a reference

Line 41: rewrite “strongly suspected by evidences”

Line 45: “swiss-army knife”

Line 87: delete common “is the consensus”

Line 101: change resume to another word

Lines 104-105: please rewrite, not clear

Line 112: should be functions

Line 113: should be domains

Line 134: delete a

Line 247: add of

Line 249: add to in, “into”

Line 257: change participating

Author Response

We are grateful to the reviewer for his/her work and for the extremely encouraging comments. We have made all the required changes and added a small paragraph of conclusions. The manuscript has been edited by a native English speaker.  

Line 24: should be threats.

OK

Lines 26-28: not sure if this necessary.

We have deleted the sentence.

Line 36: should be ends.

OK

Line 37: should be joins.

OK

Line 40: highlighted needs a reference.

Done

Line 41: rewrite “strongly suspected by evidences”

We reformulated as “These integration-independent functions are supported by evidences of several amino acid substitutions in IN – referred to as class-II mutations – affecting ….”

Line 45: “swiss-army knife”.

Done

Line 87: delete common “is the consensus”.

Done

Line 101: change resume to another word.

We have changed to “recapitulate”.

Lines 104-105: please rewrite, not clear.

Referee is right, this sentence is not clear and the concept redundant with line 40. We prefer to delete it.

Line 112: should be functions.

Ok

Line 113: should be domains.

Ok

Line 134: delete a.

ok

Line 247: add of.

Ok

Line 249: add to in, “into”

ok

Line 257: change participating.

We have changed to “associated”.